

# Hybrid quantum search with genetic algorithm optimization

Sebastian Mihai Ardelean[*] and Mihai Udrescu[*]

Department of Computer and Information Technology, University Politehnica of Timisoara, Timisoara, Timis, Romania
[*] These authors contributed equally to this work.

## ABSTRACT

Quantum genetic algorithms (QGA) integrate genetic programming and quantum computing to address search and optimization problems. The standard strategy of the hybrid QGA approach is to add quantum resources to classical genetic algorithms (GA), thus improving their efficacy (*i.e.*, quantum optimization of a classical algorithm). However, the extent of such improvements is still unclear. Conversely, Reduced Quantum Genetic Algorithm (RQGA) is a fully quantum algorithm that reduces the GA search for the best fitness in a population of potential solutions to running Grover's algorithm. Unfortunately, RQGA finds the best fitness value and its corresponding chromosome (*i.e.*, the solution or one of the solutions of the problem) in exponential runtime, $O(2^{n/2})$, where $n$ is the number of qubits in the individuals' quantum register. This article introduces a novel QGA optimization strategy, namely a classical optimization of a fully quantum algorithm, to address the RQGA complexity problem. Accordingly, we control the complexity of the RQGA algorithm by selecting a limited number of qubits in the individuals' register and fixing the remaining ones as classical values of '0' and '1' with a genetic algorithm. We also improve the performance of RQGA by discarding unfit solutions and bounding the search only in the area of valid individuals. As a result, our Hybrid Quantum Algorithm with Genetic Optimization (HQAGO) solves search problems in $O(2^{(n-k)/2})$ oracle queries, where $k$ is the number of fixed classical bits in the individuals' register.

## INTRODUCTION

Genetic algorithms (GAs) represent a widely used heuristic method for search and optimization problems inspired by evolutionary theory (*Spector, 2004*; *Matoušek, 2009*). In their simplest form—without losing generality—individuals' chromosomes encode candidate solutions as binary arrays. The GA has four phases: initialization, selection, reproduction & mutation, and termination (After initialization, the selection and reproduction & mutation phases are repeated in a loop until some condition is met, and the algorithm enters the termination phase.) In the initialization phase, the GA begins with a randomly generated population of chromosomes; the population evolves over multiple generations (each performing selection and reproduction) in search of an optimal

Corresponding author
Mihai Udrescu, mudrescu@gmail.com

solution (*Lahoz-Beltra, 2016*). Accordingly, each generation's chromosomes are evaluated on the basis of the fitness function to select the best individuals. A new generation evolves from the previous one by recombining and mutating selected individuals' chromosomes. Consequently, individuals with higher quality have a higher probability of being copied by the next generation, hence improving the population's average fitness.

However, even with sophisticated GA search strategies such as elitism or adaptive parameter control, or dedicated hardware to parallelize and accelerate GAs, classical computation often achieves only marginal performance improvements over deterministic approaches (*Spector, 2004*; *Udrescu, Prodan & Vlăduțiu, 2006*). To further pursue performance, quantum computation emerges as one of the possible GA implementation solutions due to its specific features, such as entanglement, interference, and exponential parallelism (*Nielsen & Chuang, 2002*; *Spector, 2004*). The general approach in trying to combine genetic algorithms with quantum computing is to optimize genetic operators using quantum features (*Lahoz-Beltra, 2023*); in this article, we turn the tables by proposing a classical (*i.e.,* genetic algorithm) optimization of a purely quantum search (*i.e.,* the RQGA algorithm (*Udrescu, Prodan & Vlăduțiu, 2006*)).

The remainder of this article is organized as follows: section State of the Art surveys the similar solutions to combining GAs with quantum computing, section Background describes the purely quantum RQGA search algorithm that we optimize with a genetic algorithm, section Algorithm Design details our proposed HQAGO solution to fixing qubits in the RQGA individual register and analyzes its time complexity, section Results show the results obtained by simulating HQAGO in the context of concrete optimization problems (knapsack and graph coloring), and section Conclusions discusses our findings, their implications, and potential impact. Portions of this text describing the algorithm were previously published as part of a preprint (https://doi.org/10.21203/rs.3.rs-3009060/v1).

## STATE OF THE ART

The literature proposes several quantum-implemented GAs—mostly algorithms that combine classical and quantum operators (*Lahoz-Beltra, 2016*). GAs have also been used for quantum circuit synthesis, as presented in *Ruican et al. (2007)* and *Ruican et al. (2008)*, and as evolutionary strategies that can evolve and scale up small quantum algorithms (*Gepp & Stocks, 2009*). From an implementation perspective, these trends are assembled under the term Quantum Evolutionary Programming (QEP), which largely consists of Quantum-Inspired Genetic Algorithms (QIGA) or Hybrid Genetic Algorithms (HGA). QIGAs and HGAs are algorithms that mix classical computation with quantum operators, using qubits for chromosome representations and quantum gates for operators. We have just a few examples of fully Quantum Genetic Algorithms (QGA), which focus on implementing genetic algorithms searches on quantum hardware (*Giraldi, Portugal & Thess, 2004*; *Lahoz-Beltra, 2016*).

In addition to these developments, recent optimization strategies offer promising avenues for enhancing QEP methodologies. For instance, *Escobar-Cuevas et al. (2024b)* introduces a novel method that leverages evolutionary game theory for optimization.

The proposed method initializes all individuals using the Metropolis-Hasting technique. The algorithm continuously adapts and refines the strategies of each individual based on performance—based on the interactions and the competition between individuals—in search of the global optimum or near-optimal solution. Similarly, *Escobar-Cuevas et al. (2024a)* presents a method that combines a hybrid search mechanism with the fuzzy optimization approach that shows improvements in terms of solution quality, dimensionality, similarity, and convergence criteria (*Escobar-Cuevas et al., 2024a*).

QIGAs start with generating an initial population of $n$-qubit chromosomes; then, the best solution is selected and stored by observing and evaluating the chromosomes. The algorithm evolves by performing a classical evaluation of individual chromosomes and generating a new population, using classical and quantum operators (*Giraldi, Portugal & Thess, 2004*; *Lahoz-Beltra, 2016*). QGAs also start with a population of qubit-encoded chromosomes, but the following steps use only quantum operators. A QIGA consisting of a classical genetic algorithm with quantum crossover operation applied on all chromosomes in parallel can achieve quadratic speedup over its conventional counterpart; the complexity of such a QIGA is $\mathcal{O}(\tilde{N} poly(log \tilde{N} logN))$ where $\tilde{N} \leq N$, $\tilde{N}$ is the number of individuals in a generation and $N$ is the total number of individual chromosomes (*SaiToh, Rahimi & Nakahara, 2014*). Quantum Genetic Optimization Algorithm (QGOA) is a QIGA that combines quantum selection with classical operations performing crossover, mutation, and substitution (*Malossini, Blanzieri & Calarco, 2008*). Another QIGA approach introduces a new way of implementing GA operators on quantum hardware to aim for better runtimes; however, the proposed QIGA only converges towards suboptimal solutions, and its complexity is uncertain (*Acampora & Vitiello, 2021*).

RQGA is a fully quantum genetic algorithm based on Grover's quantum search, which does not have genetic operators such as mutation and crossover. Compared to the 4-phases (initialization, fitness assessment, variation, and selection) QIGAs, the RQGA performs only initialization, fitness assessment, and selection. There is no need for a variation stage in RQGA since the individuals' register encodes the entire search space as a superposition of chromosome codes. Therefore, in the initialization phase, the population is generated as a basis-state superposition of all possible binary combinations (*Udrescu, Prodan & Vlăduțiu, 2006*). In this way, RQGA provides a solution that consists in finding the best individual/chromosome with a specially designed oracle that works with a modified version of the maximum finding algorithm (*Ahuja & Kapoor, 1999*). Overarchingly, RQGA represents a method that reduces any Quantum Genetic Algorithm (QGA) to a Grover search (*Grover, 1996*). Therefore, the complexity of RQGA is $\mathcal{O}\left(\sqrt{n_i}\right)$ Grover iterations (where $n_i$ is the number of items) in a search space with $n_i = 2^n$ items (where $n$ is the number of qubits in the search register), or $\mathcal{O}\left(2^{n/2}\right)$.

The main objective of this article is to reduce the complexity of the RQGA search by using classical optimization approaches. Consequently, the main contributions of this article are:

- A novel GA-based method of reducing the number of qubits required in the individuals' register of the RQGA. Our classical GA, combined with RQGA, or HQAGO, fixes the

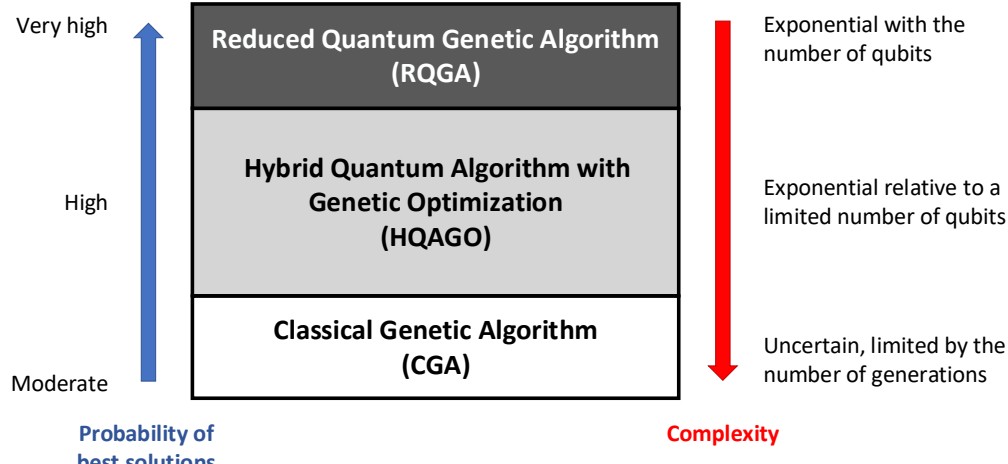

**Figure 1** **The main achievement of our HQAGO approach is that it allows for scanning the space between a pure classical GA (*i.e.*, all positions in the individuals' register are classical bits, '0' or '1') and RQGA (where the individuals' register has only qubits).** Pure classical GA has a moderate probability of finding the best solutions; however, their complexity can be restricted by the termination condition that limits the number of generations. The pure-quantum RQGA has a very high probability of finding the best solutions (according to Grover's algorithm); their complexity is exponential with the number $n$ of qubits in the individuals' register. HQAGO maintains RQGA's high probability of finding the best solutions while significantly reducing the search complexity by limiting the number of qubits in the individuals' register.

value of $k$ bits in the $n$-qubit individuals' register as classical '0' or '1', while the other register positions remain quantum (*i.e.*, qubits). Therefore, considering that Grover's algorithm delivers the complexity of the search, our HQGAO is $\mathcal{O}\left(2^{(n-k)/2}\right)$. By controlling $k$, we can control the complexity of the search so that the probability of finding a solution remains high; however, the number of required Grover iterations is reduced because a limited number of qubits means a reduced search space (see Fig. 1).

- A new method to discard unfit solutions and bound the search only in the area of valid individuals. This way, we reduce the number of Grover algorithm runs to find the best fitness value.
- Series of Qiskit simulations of HQAGO implementations for solving the *knapsack optimization* and *graph coloring* problems that show that best search solutions can be found even for relatively large $k$ values, which consequently entail a drastically reduced search space and a much lower computational complexity.

## BACKGROUND

Since HQAGO builds upon the pure-quantum RQGA, this section details the RQGA implementation and analyzes its complexity. RQGA takes a superposition of all possible individual chromosomes (representing potential solutions for the search problem) in the individuals' register $|u\rangle_{ind}$ and computes the corresponding fitness values in the fitness register $|f(u)\rangle_{fit}$. RQGA uses Grover's algorithm (*Grover, 1996*) to augment the quantum

amplitude of the basis state in $|u\rangle_{ind}$ that corresponds to the best fitness values. Thus, when we measure the fitness register $|f(u)\rangle_{fit}$, we get the best fitness value (or one of the best fitness values) with a high probability. The post-measurement state will have only the individual code (or a superposition of individual codes) that produces the best fitness. In any of these cases, measuring the individuals' register will return the solution.

RQGA is a framework built around a modified version of the quantum maximum finding algorithm proposed by *Ahuja & Kapoor (1999)*. This approach reduces the problem of finding the maximum fitness individual to a Grover search (*Udrescu, Prodan & Vlăduţiu, 2006*), which requires $\mathcal{O}\left(\sqrt{N}\right)$ Grover iterations (*Nielsen & Chuang, 2002*). RQGA encodes the search space on $n$ qubits; therefore, in our case, $N = 2^n$. Accordingly, as RQGA maintains the number of oracle queries of the quantum maximum finding algorithm, namely $\mathcal{O}\left(\sqrt{N}\right)$, RQGA's complexity becomes $\mathcal{O}\left(2^{n/2}\right)$. In Algorithm 1, we present the main steps of RQGA.

RQGA's worth is that it uses Grover's and maximum finding algorithms to simplify QGAs. However, its main drawback is that it still requires an exponential runtime; for a search space of size $2^n$, its complexity is $\mathcal{O}\left(2^{n/2}\right)$ Grover iterations (*Udrescu, Prodan & Vlăduţiu, 2006*). This situation calls for a solution to reduce or control the algorithm's complexity.

---

**Algorithm 1** The main steps of RQGA (*Udrescu, Prodan & Vlăduţiu, 2006*)

---

1: Prepare $|\psi\rangle_1$ as a superposition of all individual–fitness register pairs ($|u\rangle_{ind} \otimes |0\rangle_{fit}$), as presented in Equation 1.

2: Choose $max \in [2^{m+1}, 2^{m+2} - 1)$ randomly, where $m$ is the number of qubits in the fitness register.

3: Appy the unitary operation corresponding to the fitness function $f$: $|\psi\rangle_2 = U_{fit}|\psi\rangle_1 = \frac{1}{\sqrt{2^n}} \sum_{u=0}^{2^n-1} |u\rangle_{ind} \otimes |f(u)\rangle_{fit}$

4: **repeat**

5:      Use the oracle $\mathbb{O}$ to mark (i.e., change to a negative phase) all basis states in the fitness register that correspond to $f(u) \geqslant max$. ($|\psi\rangle_3 = \mathbb{O}|\psi\rangle_2$)

6:      Use Grover iterations to augment the quantum amplitude corresponding to the marked fitness values. Then measure the fitness register, obtaining $|\psi\rangle_4 = |u\rangle_{ind} \otimes |f(u)\rangle_{fit}$, with $f(u) \geqslant max$.

7:      $max := f(u)$.

8: **until** $max$ value is not improved.

9: Return the chromosome value $u_{max}$ (corresponding to $max$), namely $|u_{max}\rangle_{ind} \otimes |f(u_{max})\rangle_{fit}$, with $f(u_{max}) = max$. Therefore, $u_{max}$ represents the individual/chromosome that generates the highest fitness.

---

# ALGORITHM DESIGN

In this article, we reduce the RQGA exponential runtime by limiting the number of qubits in the search register. Our novel Hybrid Quantum Algorithm with Genetic Optimization

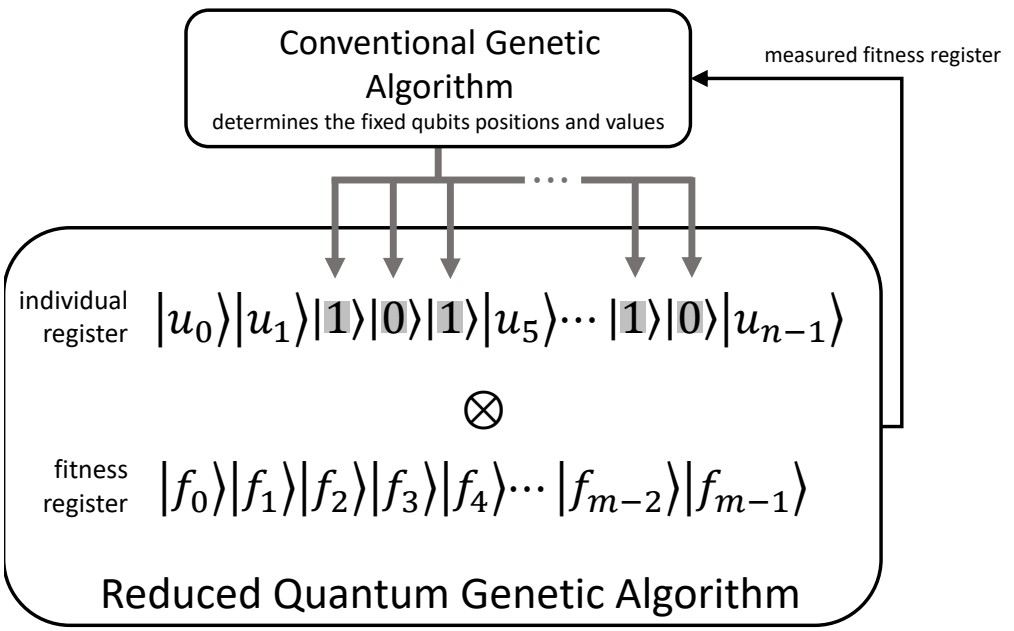

**Figure 2** **The overview of applying genetic algorithm optimization to reduce the number of Grover iterations entailed by running the RQGA algorithm.** The conventional genetic algorithm determines the fixed qubits' positions (presented with gray background) and their binary values in the individuals' register of the Reduced Quantum Genetic Algorithm, thus controlling the number of qubits in the individual/chromosome quantum register and reducing the number of Grover iterations required.

(HQAGO) algorithm selects a bounded number of qubits in the individuals' register and fixes the remaining ones as classical values of '0' and '1'; a classical genetic optimization algorithm selects the qubits' positions and determines the values of the fixed bits. Compared to RQGA, where the population contains both valid and non-valid individuals, HQAGO also modifies the initialization step to search only in the valid individuals' space. (An individual chromosome is valid if it meets a condition specific to the search or optimization problem; it is non-valid otherwise.) With the HQAGO procedure presented in Fig. 2, we reduce the number of Grover iterations, thus improving the algorithm's performance, at the cost of adding complexity—entailed by genetic optimization—to the RQGA design.

Like RQGA, the HQAGO starts by initializing a superposition of all individual-fitness register pairs as

$$|\psi\rangle 1 = \frac{1}{\sqrt{2^n}} \sum_{u=0}^{2^n-1} |u\rangle_{ind} \otimes |0\rangle_{fit}, \tag{1}$$

where $|u\rangle_{ind} \otimes |0\rangle_{fit}$ is the individual-fitness register pair and $n$ is the number of qubits in the individuals' quantum register. The individual is encoded on $n$-qubits; therefore we have $2^n$ basis states in the superposition.

Given the individual quantum register $|u\rangle_{ind}$ we apply the classical GA to fix a subset of $k$ qubits (*i.e.*, assign them classical values of 0 and 1), $0 \leq k \leq n$. Before fixing qubits in the individuals' register, in Equation Eq. (1) we have $|u\rangle \in S = \{0, 1, 2, \dots, 2^n - 1\}$; $u \in \mathbb{N}$ is

binary-encoded ($u = b_0 b_1 \ldots b_{n-1}$ where $b_i \in \mathbb{B} = \{0,1\}$, $i = \overline{0, n-1}$). When we assign the classical binary values to $k$ $b_i$s, $|u\rangle \in S_k \subseteq S$; the cardinality of $S_k$ is $|S_k| = 2^{n-k}$ elements. For example, for $n = 4$ and $k = 2$, we have $u = b_0 b_1 b_2 b_3$, and fix $b_1 = 1$ and $b_2 = 0$; in this case, $S_k = \{4, 5, 12, 13\}$ (in binary, $\{0100, 0101, 1100, 1101\}$, where the circled bits are fixed). Thus, we obtain the next state

$$|\psi\rangle 2 = GA|\psi\rangle 1 \longmapsto \frac{1}{\sqrt{2^{n-k}}} \sum_{u \in S_k} |u\rangle_{ind} \otimes |0\rangle fit. \tag{2}$$

We present the complete initialization phase in Algorithm 2, and the conventional GA chromosome initialization in Algorithm 3.

---

**Algorithm 2** HQAGO initialization, identical with RQGA

---

1: Initialize the $n$-qubits individual quantum register $|u\rangle = |0\rangle^{\otimes n}$
2: Initialize the $(m+1)$-qubits fitness quantum register $|fitness_u\rangle = |0\rangle^{\otimes(m+1)}$
3: Initialize the oracle workspace 1-qubit quantum register $|ws\rangle = |0\rangle$
4: Create the quantum circuit $QC$
5: Apply conventional GA such that $GA_{solution} = GA(|u\rangle)$, where $GA_{solution}$ encodes the values and the positions of the fixed $k$ qubits.
6: $|u\rangle = H^{\otimes n}|0\rangle^{\otimes n} \longmapsto \frac{1}{\sqrt{2^n}} \sum_{u=0}^{2^n-1} |u\rangle$
7: $|ws\rangle = H|0\rangle \longmapsto \frac{1}{\sqrt{2}}(|0\rangle + |1\rangle)$

---

**Algorithm 3** Conventional genetic algorithm individual initialization

---

1: **while** gene is not generated **do**
2:     Generate a random value that represents the gene value.
3:     **if** gene is not already generated **then**
4:         Randomly generate the sign of the gene.
5:         **if** sign is 0 **then**
6:             Gene value is negated.
7:         **end if**
8:         Append gene value.
9:     **end if**
10: **end while**

---

The conventional GA searches for the optimal configurations that maximize the fitness, given the search space limitations dictated by fixing qubits in the individuals' register. The GA is a classical (*i.e.,* non-quantum) algorithm that starts by generating an initial population according to Algorithm 3 and then calculates the fitness for each chromosome. To define a format that encodes each fixed qubit's value and position in the register, we define a constraint on the chromosome format. As such, we consider that the absolute value of the gene $v$ encodes the position of the fixed qubit; the sign of the gene encodes the fixed qubit's value. Therefore, a negative $v$ means '0' on position/index $v$ in the individual

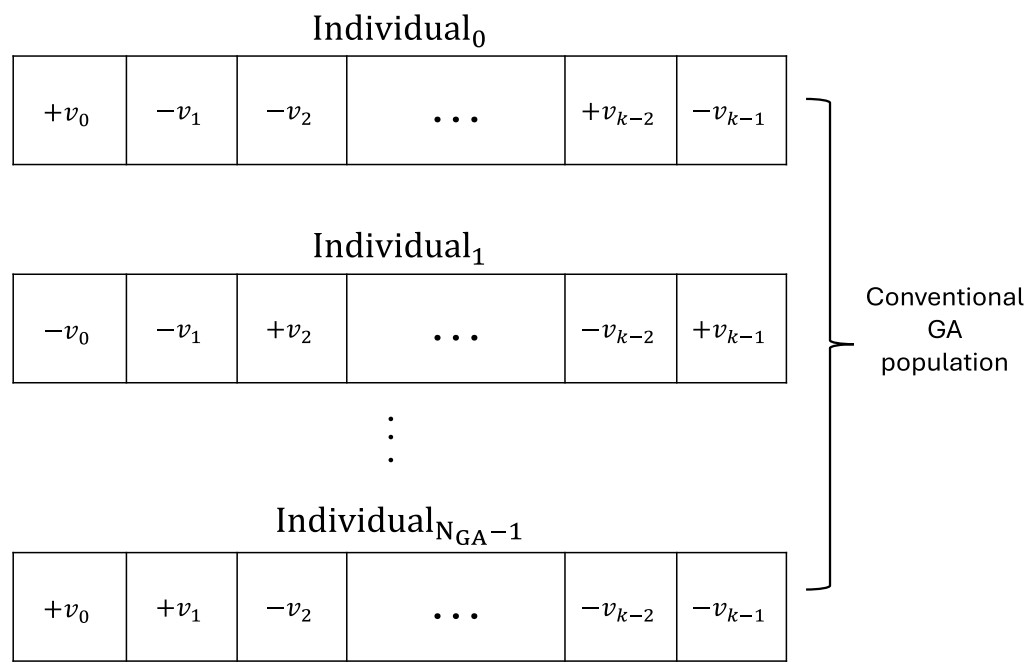

**Figure 3** **An example of chromosome encoding.** The absolute value of the gene $v$ encodes the position of the fixed qubit while the gene encodes the fixed qubit's value. $-v_i$, $i = \overline{0, k-1}$ means '0' on position $v_i$ in the individual quantum register, while $+v_i$ means '1' on position $v_i$ in the individual quantum register. $N_{GA}$ is the number of individuals in the conventional GA's population.

quantum register ($b_v = 0$), while a positive $v$ means '1' on position $v$ in the individual quantum register ($b_v = 1$). In Fig. 3 we present an example of chromosome encoding in the conventional GA's population.

The classical GA evolves the population of chromosomes across multiple generations in search of the maximum fitness (which corresponds to the solution). Each generation of chromosomes is evaluated to select the fittest individuals; we used a probabilistic method where the chances of being selected are proportional to the respective fitness values (*Spector, 2004*). The percentage of the population selected for crossover is 32% (similar to other classical GA approaches) (*Stanhope & Daida, 1998*). Then, we perform fixed point crossover and random mutation (with an adaptive mutation rate) to obtain a new generation of offspring chromosomes. (We did not use elitism for the fittest individuals.) The termination conditions are met when, as shown in Algorithm 4, we find an optimal solution (corresponding to the maximum fitness) or the number of generations exceeds a maximum number (which is given as a parameter). In Fig. 4A, we present the conventional GA operator symbol that we integrate in the HQAGO design, while Fig. 4B presents the circuit implementation of the operator.

The next step in HQAGO is to calculate the superposed fitness values of all individuals in the fitness register. Such a quantum fitness function maintains the correlation between each individual and its corresponding fitness value; it is applied to valid and non-valid individuals. Thus, as presented in *Udrescu, Prodan & Vlăduţiu (2006)* the

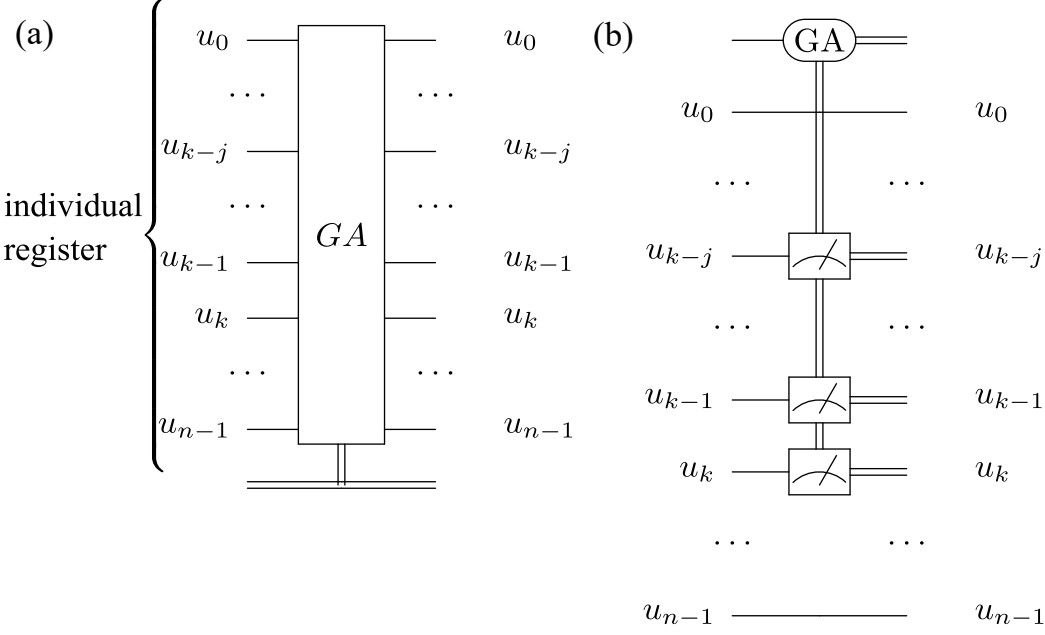

**Figure 4** **Classical GA circuit applied on the individuals' quantum register.** (A) We present the GA operator symbol while in (B) we present the gate-level implementation of the circuit.

---

**Algorithm 4** Conventional Genetic Algorithm optimization

---

1: **for each** individual in population **do**

2:     Initialize *individual*.

3:     Calculate fitness.

4: **end for**

5: Select fittest individuals from *population*.

6: **while** fitness < maximum fitness and maximum number of generations not exceeded **do**

7:     Save the fittest individuals (selection) in order to form the new population.

8:     Apply crossover operation on selected individuals and save the offsprings.

9:     Mutate the new population resulting from the fittest individuals and offsprings.

10:     Select the fittest individuals from the new population.

11: **end while**

---

assessment operator $U_{fit}$, is a unitary operator characterized by a Boolean fitness function $f_{fit} : \{0,1\}^n \rightarrow \{0,1\}^{m+1}$,

$$f_{fit}(x) = \begin{cases} 0 \times \{0,1\}^m, & x \text{ is a non-valid individual} \\ 1 \times \{0,1\}^m, & \text{otherwise,} \end{cases} \tag{3}$$

where $m$ represents the number of qubits in the fitness register.

The fitness value is encoded using $(m+1)$-qubits with the most significant one indicating the validity of the individual; when the most significant bit is '0', it means a non-valid

individual; when '1', it means a valid one. As such, the values returned by $f_{fit}$ represented in two's complement belong to distinct fitness areas corresponding to valid and non-valid individuals (a non-valid chromosome configuration represents a combination that does not satisfy some given conditions) (*Udrescu, Prodan & Vlăduţiu, 2006*). Naturally, $U_{fit}$ characterized by the fitness function $f$ is an unitary operator,

$$U_{fit} : |u\rangle_{ind} \otimes |0\rangle_{fit} \longmapsto |u\rangle_{ind} \otimes |ffit(u)\rangle, \tag{4}$$

where $|u\rangle_{ind} \otimes |\bullet\rangle_{fit}$ is the individual-fitness value quantum pair register($|\bullet\rangle$ stands for either $|0\rangle$ or $|f_{fit}(u)\rangle$).

Explicitly applying the $U_{fit}$ operator on all superposed individuals means

$$|\psi\rangle 3 = U_{fit}|\psi\rangle 2 = U_{fit} : \frac{1}{\sqrt{2^{n-k}}} \sum_{u \in S_k} |u\rangle_{ind} \otimes |0\rangle_{fit} \longmapsto \frac{1}{\sqrt{2^{n-k}}} \sum_{u \in S_k} |u\rangle_{ind} \otimes |f_{fit}(u)\rangle_{fit}. \tag{5}$$

In Fig. 5, we present the symbol of the $U_{fit}$ operator, with input and output qubits; Fig. 6 shows the gate-level implementation of the operator. Algorithm 5 explains the assessment by fitness operation.

---

**Algorithm 5** Assessment operation

| | |
|---|---|
| 1: | **for each** individual in population **do** |
| 2: | Calculate fitness |
| 3: | Apply $U_{fit}$ operator |
| 4: | **if** fitness value is valid **then** |
| 5: | Mark individual as valid by setting $f_M = 1$. |
| 6: | **end if** |
| 7: | **end for** |

---

In the next step, we apply the Oracle and Grover diffuser (*i.e.*, the Grover iteration) $\mathcal{O}\left(\sqrt{2^{(n-k)}}\right)$ times. Like in RQGA, we generate a random value $max \in \mathbb{N}, max > 0$ in the interval $[2^{m+1}, 2^{m+2} - 1)$, such that the search for the individual with the highest fitness will occur in the valid individuals' area (*Udrescu, Prodan & Vlăduţiu, 2006*). The oracle $\mathbb{O}$ operates on the fitness quantum register qubits except for the validity qubit $v$ (see Fig. 7), and uses two's complement representation for marking the states with a value greater than $max$. (By subtracting $max$ from all fitness values, only the fitnesses equal or greater than $max$ will remain positive and will be marked with a negative phase.)

Accordingly, the oracle $\tilde{\mathbb{O}}_{max}\left(f_{fit}(u)\right)$ is applied on the register $|\bullet\rangle_{fit}$ from state $|\psi\rangle_3$,

$$|\psi\rangle_4 = \tilde{\mathbb{O}}_{max}|\psi\rangle 3 \longmapsto (-1)^{g(u)} \frac{1}{\sqrt{2^{n-k}}} \sum_{u \in S_k} |u\rangle_{ind} \otimes |ffit(u)\rangle fit, \tag{6}$$

where

$$g(u) = \begin{cases} 1 & if \quad |f_{fit}(u)\rangle_{fit} \geqslant max \\ 0 & otherwise. \end{cases} \tag{7}$$

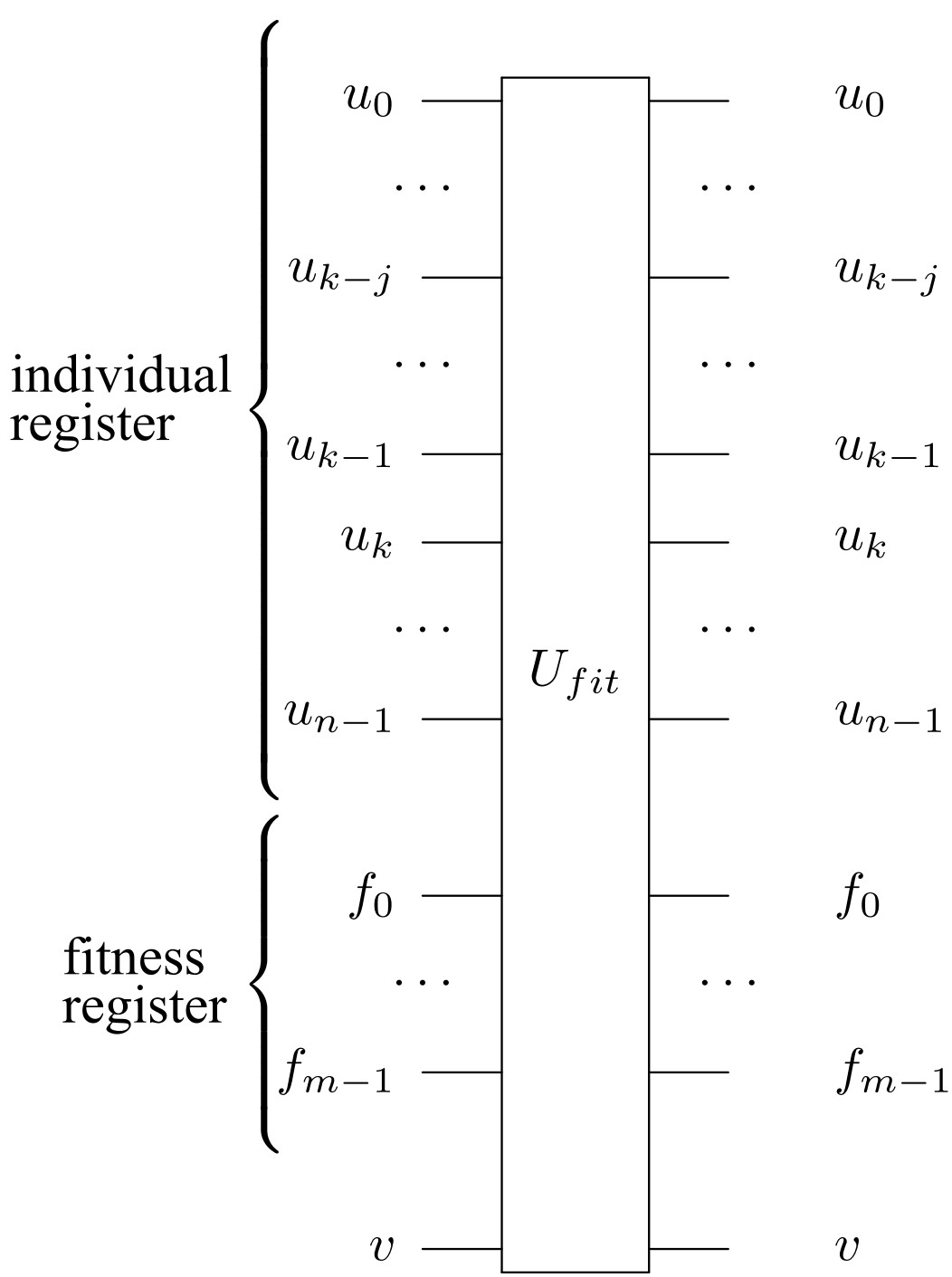

**Figure 5** The symbol of the $U_{fit}$ circuit, its inputs and outputs.

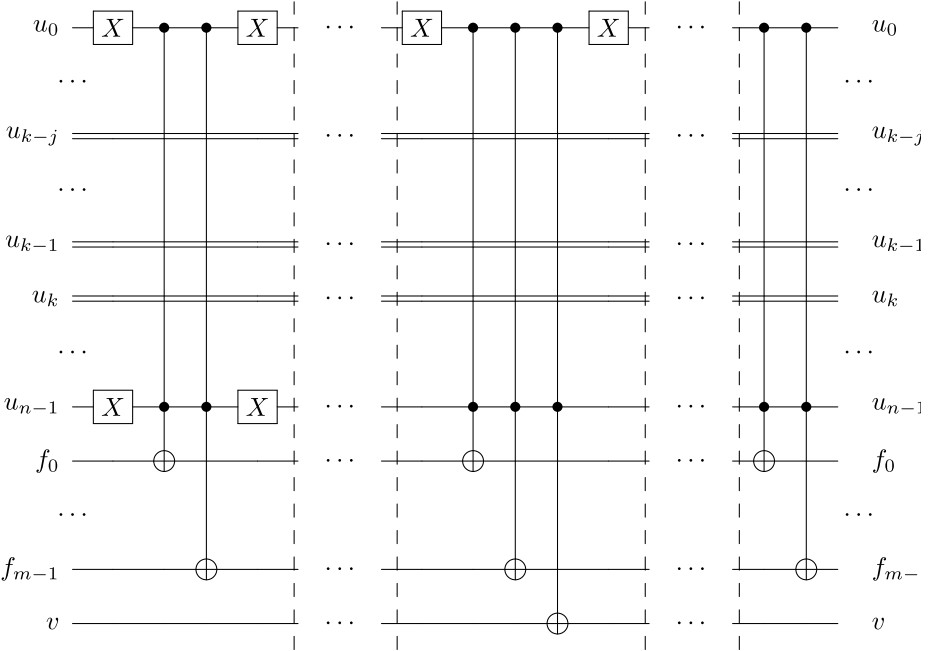

**Figure 6** **The gate-level implementation of the $U_{fit}$ sub-circuit utilizes $n$-qubit Controlled-not gates** (***Nielsen & Chuang, 2002***). The qubits from the individuals' register are control qubits, and the qubits from the fitness registers are target qubits. $v$ is the valid qubit that indicates the validity of the corresponding chromosome.

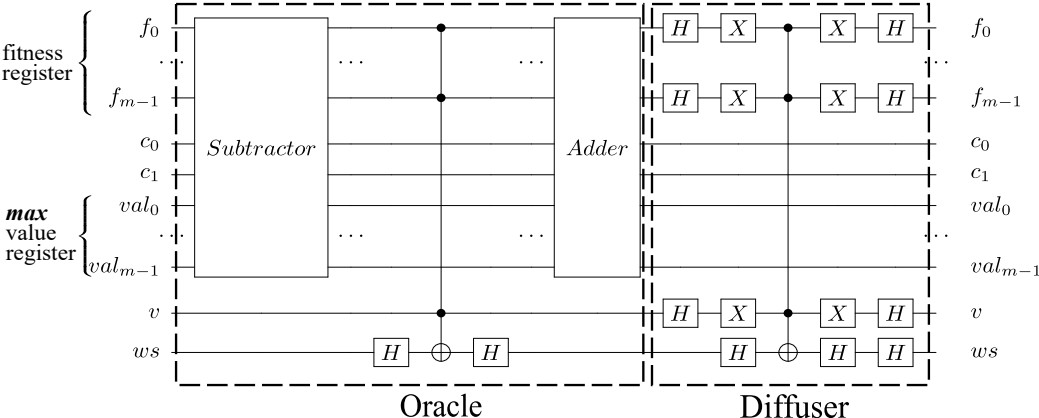

**Figure 7** **Grover circuit.** The oracle uses 2 two's complement quantum adders, 2 Hadamard gates, and 1 $n$-qubit Controlled-not gate. *Max value register* is the quantum register storing the *max* value, while $c_0$ and $c_1$ are the carry qubits used in the subtraction and addition circuits; $v$ is the valid qubit that indicates the validity of the corresponding chromosome; *ws* is the oracle workspace qubit (***Udrescu, Prodan & Vlăduțiu, 2006***). The diffuser utilizes Hadamard, Pauli-X, and $n$-qubit Controlled-not gates.

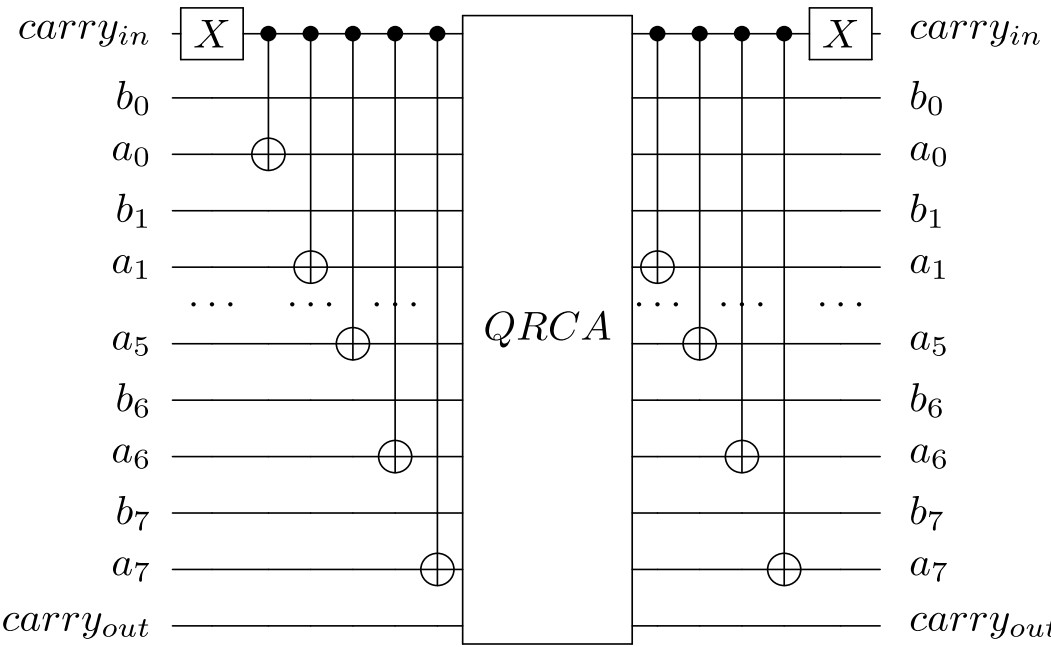

**Figure 8** Quantum subtractor design on 8-qubit numbers using QRCA as in (*Cuccaro et al., 2004*), where $a_0, a_1, \ldots, a_7$ represent the first operand and $b_0, b_1, \ldots, b_7$ encode the second operand.

The oracle $\mathbb{O}$ is implemented using two's complement quantum adders and subtractors (*Udrescu, Prodan & Vlăduţiu, 2006*); it is applied on the entire fitness register, except for the validity qubit. Using two's complement addition does not affect the correlation between the individual and its corresponding fitness value since addition is a pseudo-classical permutation function. Hence, by subtracting and adding $max + 1$ to the fitness register, all basis states for which the fitness value is greater than $max + 1$ are marked by multiplying their amplitudes with $-1$. (In other words, marking a superposed state means its amplitude becomes negative.)

We may consider Quantum Carry Look-Ahead Adder (QCLAA), as presented in *Cheng & Tseng (2002)*, or Quantum Ripple Carry Adder (QRCA), see *Cuccaro et al. (2004)*, as possible implementations for the quantum adders. Figure 8 presents the gate-level implementation of the subtractor using QRCA. We opted for a ripple-carry adder because it offers an advantage over the Quantum Carry Look-Ahead Adder (QCLAA) in terms of the number of qubits used. For an $n$-qubits individuals' register and $\mathcal{O}\left(\sqrt{2^{n-k}}\right)$ Grover iterations, using the QRCA circuit requires $2\sqrt{2^{n-k}} + 1$ carry qubits (1 carry-in qubit and 2 qubits for carry-out in each iteration—1 carry-out qubit for each adder). The QCLAA requires a total of $2(n-k+1)$ carry qubits in each iteration, namely $n-k+1$ carry qubits for each adder. Therefore, from the perspective of the additional required qubits, using QCLAA is not an acceptable solution for our implementation.

Next, we iterate the Grover diffuser $\sqrt{2^{n-k}}$ times, to augment the amplitudes of the marked states $|\psi\rangle i = |f_{fit}(u)\rangle i$ with $f_{fit}(u) \geqslant max$ in the fitness register; thus, the resulting

population becomes

$$|\psi\rangle 5 = \mathbb{G}|\psi\rangle_4. \tag{8}$$

In Algorithm 6 we present the effects of using the Grover circuit implemented according to Fig. 7, where $|ws\rangle$ represents the workspace.

---

**Algorithm 6** Grover algorithm

---

1: Subtract *max* value from the fitness values
2: $|ws\rangle = H|ws\rangle$
3: $|ws\rangle = CNOT(|fitness_u\rangle, |ws\rangle)$
4: $|ws\rangle = H|ws\rangle$
5: Add *max* value to the fitness value
6: Use Grover iteration to find the marked states, $|\psi\rangle = |f_{fit}(u)\rangle_i$ with $f_{fit}(u) \geqslant max$, in the fitness register.

---

After iterating the Grover diffuser $\sqrt{2^{n-k}}$ times, we measure the fitness register $|\bullet\rangle$ to obtain (with a high probability) a fitness value $\geq max$ in $|\bullet\rangle$; thus, in the individual register, we get a superposition of individuals that generate fitness values $\geq max$. We then update the *max* value with the measured fitness value. The entire Grover algorithm procedure is applied multiple times until the *max* value is no longer improved, and the measured fitness value corresponds to the solution (or one of the solutions). To find the solution that solves our problem, we need to measure the individual register (in this state, the individual register is a superposition of individuals that generate the highest fitness). Algorithm 7 presents the entire implementation of our HQAGO method, and in Fig. 9 the circuit implementation. (In Supplemental Information, Knapsack problem example, we present a step-by-step example of how Algorithm 7 works on an instance of the Knapsack problem.)

**Space complexity**

Solving real-world problems using quantum algorithms requires large numbers of qubits when accounting for error correction. As mentioned in (*Tănăsescu, Constantinescu & Popescu, 2022*), factoring a 2,048-bit number using Shor's algorithm (*Shor, 1994*) requires 400,000 qubits when error correction is accounted for. In our previous work, see (*Ardelean & Udrescu, 2022a*), we showed that for solving the knapsack problem, the total number of qubits required by RQGA grows exponentially as the number of qubits used for individual representation grows. Thus, for solving real-world problems using error-corrected qubits is necessary to implement hybrid solutions that capitalize on the quantum speed advantage and reduce the number of required logical qubits.

Our HQAGO requires $n$ qubits to encode the individuals' register, $m+1$ qubits for the fitness register, and $m$ qubits for the *max* value representation. Additionally, the algorithm requires $2 \cdot r + 1$ carry-qubits in the oracle architecture and one qubit for the Oracle workspace. Altogether, the space complexity of the algorithm is $2 \cdot (m+1+r) + n + 1$.

In Fig. 10, we compare the pure quantum configuration of HQAGO (*i.e.,* no fixed qubits) with the algorithm configurations with 2 and 3 fixed qubits. The circuit's critical

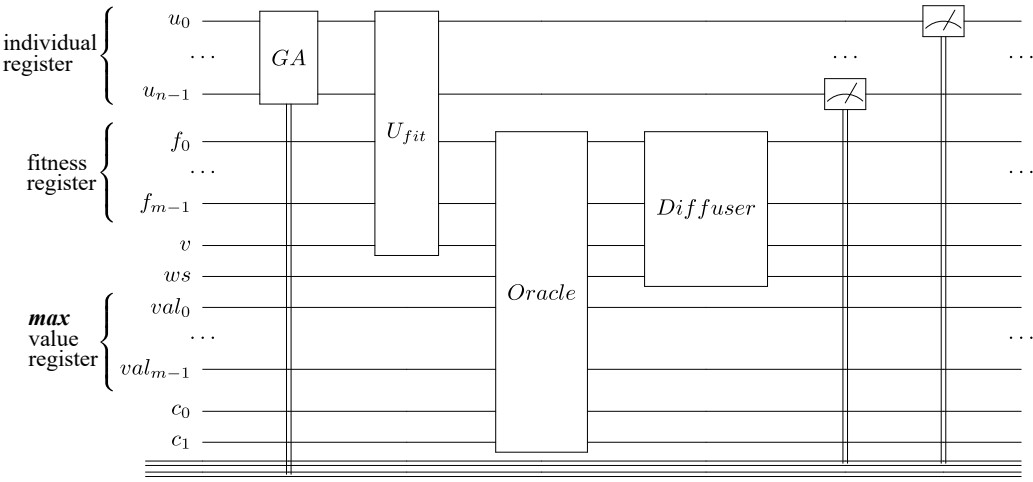

**Figure 9** **Hybrid quantum algorithm with genetic optimization circuit implementation.** The $u$ qubits make out the individuals' quantum register, $f$ qubits represent the fitness quantum register, while $v$ is the valid qubit; the *val* qubits represent the *max* value (*Udrescu, Prodan & Vlăduţiu, 2006*). The carry-in and carry-out qubits used by adder sub-circuits are $c_0$ and $c_1$. For simplicity, we represent only one Grover Iteration and one maximum finding iteration.

---

**Algorithm 7** Hybrid Quantum Algorithm with Genetic Optimization

---

1: $|\psi\rangle_1 = \frac{1}{\sqrt{2^n}} \sum_{u=0}^{2^n-1} |u\rangle_{ind} \otimes |0\rangle_{fit}$.

2: Apply conventional Genetic Algorithm outcome outcome, $|\psi\rangle_2 = GA|\psi\rangle_1$.

3: Apply unitary operation $U_{fit}$ corresponding to fitness computation, $|\psi\rangle_3 = U_{fit}|\psi\rangle_2$

4: Randomly generate the real value $max \in [2^{m+1}, 2^{m+2} - 1)$

5: **repeat**     ▷ Iterates $N_{mf}$ times, where $N_{mf}$ represents the number of iterations of the maximum finding algorithm.

6:     Apply the oracle $\mathbb{O}$ on the entire fitness register except for the validity qubit. $|f_{fit}(u)\rangle_{fit}$ basis states are marked if $|f_{fit}(u)\rangle_{fit} \geqslant max$.

7:     Use Grover iteration to find the marked states, $|\psi\rangle = |f_{fit}(u)\rangle_{fit}$ with $f_{fit}(u) \geqslant max$, in the fitness register.

8:     $max = |\psi\rangle$.

9: **until** until $max$ no longer improves.

10: Measure $|u\rangle_{ind}$ register in order to obtain the corresponding individual which represents the solution.

---

path length is the same in all three setups. As shown, the complexity of the quantum circuit decreases as we increase the number of fixed qubits.

## Time complexity

HQAGO fixes $k$ qubits, meaning that the time complexity of the quantum part is $N_{gi} \times N_{mf}$, where $N_{gi} = \sqrt{2^{n-k}}$ is the number of Grover iterations and $N_{mf}$ is a linear function of $n$ that represents the number of iterations of the maximum finding algorithm (*Ahuja & Kapoor, 1999*). This way, we control the time complexity of HQAGO by increasing $k$.

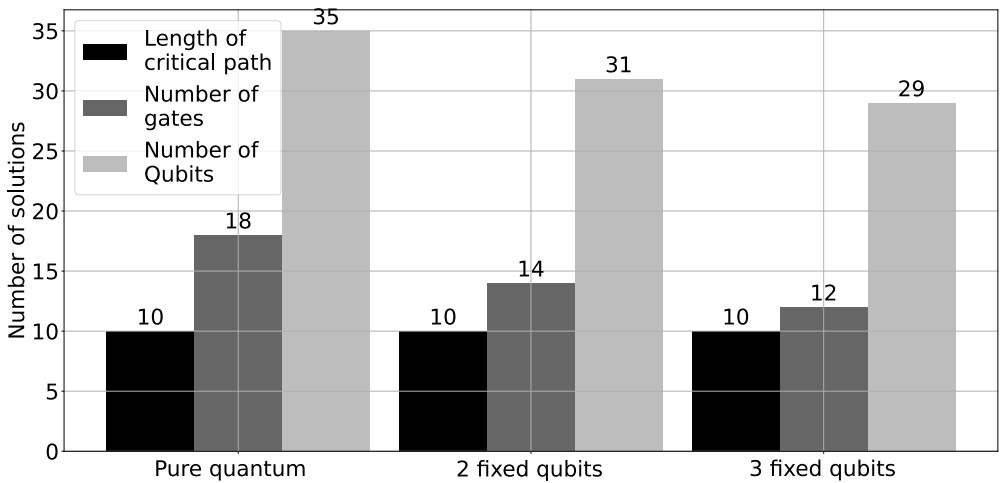

**Figure 10** **Comparison between the pure quantum HQAGO (equivalent to RQGA) and the HQAGO with 2 and 3 fixed qubits, from the perspective of circuit complexity.** (The critical path length is the maximum number of gates between the input and the output in the quantum circuit.).

However, the total time complexity of HQAGO comprises both the quantum and the classical GA parts. The time performance of the classical GA depends on the application domain and implementation parameters (*Ankenbrandt, 1991*); it can be predicted as a function of the population size, cardinality of the representation, the complexity of the evaluation function, and the fitness ratio. Nonetheless, the assessment of convergence in the classical GA is beyond the scope of this study. Still, we note that bounding the number of generations controls the classical GA runtime.

Indeed, HQAGO aims to reduce the algorithm's complexity by reducing the number of Grover iterations, thus improving the performance at the cost of adding the classical GA. For the individual quantum register $|u\rangle_{ind} \in S, |S| = 2^n$ we fix a subset of $k$ qubits—using classical GA—such that $|u\rangle_{ind} \in S_k \subseteq S, |S_k| = 2^{n-k}$. Therefore, we employ Grover's search algorithm (*Grover, 1996*) on a reduced search space $S_k$, so that HQAGO requires $\mathcal{O}\left(\sqrt{2^{n-k}}\right)$ oracle queries for NP-hard problems with unique solution (global optimum), and $\mathcal{O}\left(\sqrt{\frac{2^{n-k}}{M}}\right)$ for problems with $M$ solutions (*Nielsen & Chuang, 2002*). The best performance of HQAGO is determined experimentally by finding the "sweet spot" in which the number of classical GA generations and the number of Grover iterations is minimized. In Fig. 11 we present the complexity reduction both theoretical (according to the $\mathcal{O}$ notation functions) and simulated. As shown, the algorithm's complexity decreases exponentially as the number of fixed qubits increases.

## RESULTS

We use the Qiskit toolchain (*Javadi-Abhari et al., 2024*) to analyze the conventional GA's convergence and measure the quantum algorithm's performance. Qiskit is an open-source library for quantum computing that enables interaction with the IBM Q hardware and

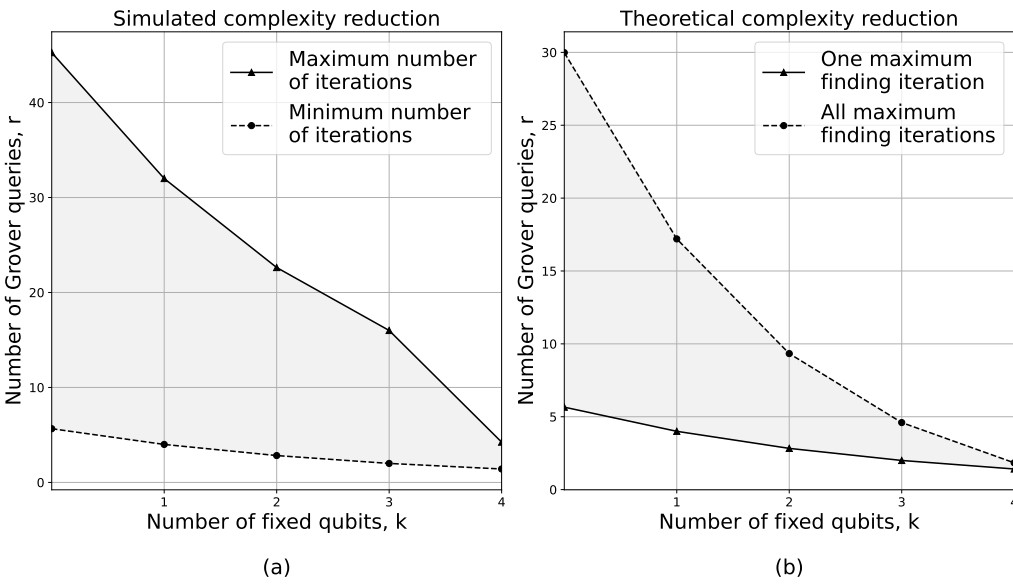

**Figure 11** **(A) Presents the theoretical complexity reduction of HQAGO according to the calculated** $\mathcal{O}$**-notation formulas. We calculate the number of iterations for the maximum finding algorithm (** $N_{mf}$ **) using a linear function** $N_{mf}(x) = a \cdot x + b$ **where** $x$ **is the number of qubits, and** $a = 1$ **and** $b = 0.3$ **are the fixed-values parameters approximated after multiple experiments. In (B) we present the complexity reduction of the algorithm determined after simulating the knapsack problem.**

fosters the development and simulation of quantum algorithms (*Wille, Van Meter & Naveh, 2019*). We instantiated HQAGO, as presented in Fig. 9, using the **IBMQ** back end, **simulator_mps** (version 0.1.547 with a configuration of 16 shots) from the **ibm-q** provider. The simulator is a tensor-network simulator that uses Matrix Product State representation—limited to 100-qubit circuits. The following basic gates are available on **simulator_mps**: U1, U2, U3, U, P, CP, CX, CZ, ID, X, Y, Z, H, S, SDG, SX, T, TDG, SWAP, CCX, UNITARY, ROERROR, DELAY.

To assess HQAGO performance, we propose two applications representing instantiations of the algorithm: one that solves the knapsack problem and the second one solves graph coloring problems. We also compare the outcome of the graph Coloring problem simulation with our previous results (*Ardelean & Udrescu, 2022b*).

## Knapsack problem

The knapsack problem is defined as the task to efficiently fill a fixed capacity knapsack with items from a finite set. Let $W$ denote the maximum weight the knapsack can accommodate and $T$ the total number of available items; $w_i$ represents the weight of the $i$-th item, and $p_i$ represents its value. The goal is to load the knapsack in a way that maximizes the total value of the items while keeping the weight within the capacity limit.

The knapsack problem is a well-studied NP-hard problem with numerous applications in fields such as machine scheduling, space allocation, asset optimization, financial modeling, production and inventory management, design of network models, and traffic overload control in telecommunication systems (*Badiru, 2009*; *Bretthauer & Shetty, 2002*). Other

applications focus on scheduling hard real-time tasks and deterministic cache memory utilization (*Nawrocki et al., 2009*).

We consider a knapsack with a maximum capacity $W = 20$ kilograms and the following $T = 5$ items: $Item_0$ has 3 kg and a value of 3\$, $Item_1$ has 2 kg and a value of 5\$, $Item_2$ has 4 kg and a value of 10\$, $Item_3$ has 7 kg and a value of 5\$, and $Item_4$ has 9 kg and a value of 15\$. Therefore, we define a search space of 5 qubits, each representing an individual. We then variate the number of fixed individuals: from 0 (representing a pure quantum solution) to 5 (representing a classical GA). We perform each simulation 100 times and record the number of solutions found—in terms of local and global maximums—and the average number of Grover iterations and classical GA generations. Under these experimental conditions, we performed 3 types of experiments, implementing distinct strategies for mutation in the conventional GA that fixes qubits in the individuals' register: GA with non-adaptive mutation, GA with adaptive mutation probability, and GA with the adaptive percentage of the mutated genes. In *Supplementary Information, Knapsack problem example*, we present an example of how to apply HQAGO to solve the Knapsack problem.

We configured the classical GA algorithm to use roulette-wheel selection, single-point crossover, and random mutation. The crossover probability is 0.6, 2 parents are involved in the crossover, and the mutation rate for the non-adaptive mutation is 0.00002. We configured a population of 100 individuals that would evolve over 100 generations, with the possibility to stop the evolution after a saturation point of 30 generations. For the experiments in which we use adaptive mutation probability, the individual with the worst fitness has a 0.15 probability of mutation; in contrast, the individual with the best fitness has a probability of 0.005. We mutate 21% of the genes of the individual with the worst fitness and 13% of the genes of the individual with the best fitness in the simulations in which we use mutation with the adaptive percentage of the mutated genes. (We adopted these GA parameter values inspired by previous approaches in using GAs for quantum circuit synthesis (*Ruican et al., 2008*).)

In Supplemental Information, Conventional GA with non-adaptive mutation, Fig. S1, the pure quantum HQAGO finds the best solution after 8 RQGA iterations, while in Figs S2 and S3, S4, and S5 we notice that the number of iterations decreases. Thus, using classical GA to fix genes reduces the number of RQGA (HQAGO with no fixed qubits) iterations because non-valid solutions are discarded. In Fig. S6 from Supplemental Information, Conventional GA with non-adaptive mutation, we present the results—in terms of best and valid solutions—of the HQAGO with all the genes fixed (representing a classic GA). As presented, the best outcome is achieved after 21 classical GA generations.

We achieved the same expected outcome after using adaptive mutation for the classical GA. In Figs. S7, S8, and S9 from Supplemental Information, Conventional GA with adaptive mutation probabilities, we show that HQAGO finds the best outcome after eight RQGA iterations. Moreover, by fixing more genes, we significantly decrease the number of iterations, as presented in Figs. S10 and S11. As illustrated in Fig. S12, the classic HQAGO (*i.e.,* all qubits in the individuals' register are fixed) requires 25 classical GA generations to find the best outcome.

**Table 1** **Summary of results from the plots presented in the Supplemental Information, Knapsack problem.** The table shows the number of fixed qubits, GA generations, RQGA generations, best solutions (global maximums), and valid solutions (local maximums) for different ga configurations.

| | Number of fixed qubits | Number of GA generations | Number of RQGA generations | Number of valid solutions | Number of best solutions |
|---|---|---|---|---|---|
| Conventional GA with non-adaptive mutation | 0 fixed qubits (pure quantum solution) | 0 | 8 | 79 | 19 |
| | 1 fixed qubit | 10 | 8 | 78 | 21 |
| | 2 fixed qubits | 10 | 8 | 52 | 43 |
| | 3 fixed qubits | 10 | 6 | 53 | 34 |
| | 4 fixed qubits | 10 | 2 | 62 | 28 |
| | 5 fixed qubits (classical GA) | 21 | 0 | 62 | 38 |
| Conventional GA with adaptive mutation probabilities | 0 fixed qubits (pure quantum solution) | 0 | 8 | 79 | 19 |
| | 1 fixed qubit | 10 | 8 | 74 | 25 |
| | 2 fixed qubits | 10 | 8 | 59 | 36 |
| | 3 fixed qubits | 10 | 6 | 48 | 43 |
| | 4 fixed qubits | 10 | 2 | 64 | 29 |
| | 5 fixed qubits (classical GA) | 22 | 0 | 56 | 41 |
| Conventional GA with adaptive percentage of mutated genes | 0 fixed qubits (pure quantum solution) | 0 | 8 | 80 | 20 |
| | 1 fixed qubit | 10 | 8 | 82 | 16 |
| | 2 fixed qubits | 10 | 8 | 56 | 41 |
| | 3 fixed qubits | 10 | 6 | 37 | 46 |
| | 4 fixed qubits | 10 | 2 | 65 | 24 |
| | 5 fixed qubits (classical GA) | 21 | 0 | 61 | 35 |

Changing the percentage of the mutated genes adaptively, the algorithm (as presented in Figs. S14, S15, S16, and S17 from the Supplemental Information, Conventional GA with adaptive percentage of the mutated genes) requires fewer RQGA iterations than the pure quantum solution (see Fig. S13) or the classic HQAGO (all individual qubits fixed, see Fig. S18). In Table 1 we show a summary of the results presented in Supplemental Information, Knapsack Problem.

As presented in Figs. 12A, 12C, and 12E, the average number of Grover iterations decreases as we increase the number of fixed qubits. The experiment confirms our expectations that, by using classical GA to fix genes, the search space size is reduced (our search space is represented only by valid solutions while non-valid ones are discarded). Therefore, our approach reduces the complexity of the quantum search algorithm. The average number of Grover iterations—calculated as the product between the number

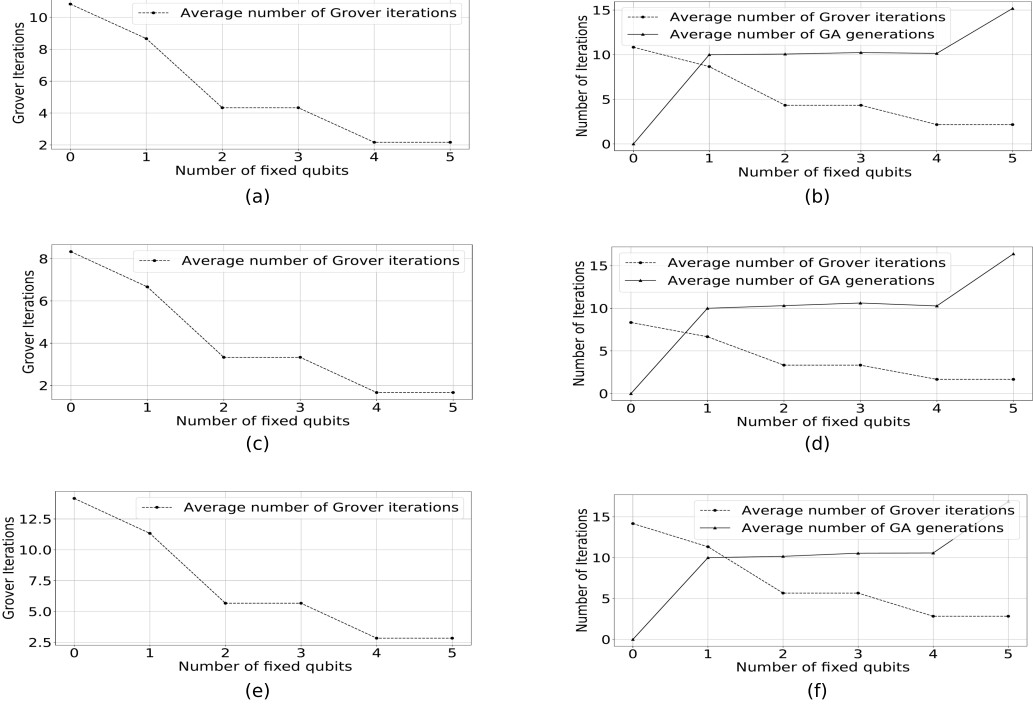

**Figure 12** **(A, C, and E) show the number of Grover iterations to find the solution for the Knapsack problem with $m = 8$ and $n = 5$ while in (B, D, and F) we present the relationship between the average number of Grover iterations and the average number of GA generations.** (A, B) The conventional GA has non-adaptive mutation. In (C, D) the conventional GA has adaptive mutation probability, while (E, F) have adaptive percentages of the mutated genes.

of Grover iterations per RQGA iteration and the average number of RQGA iterations—decreases as the search space is reduced by fixing genes. In Table 2 we summarize the results presented in Figs. 12A, 12C, and 12E.

In Figs. 12B, 12D, and 12F we present the relationship between the average number of Grover iterations and the average number of classical GA generations. By variating the number of fixed qubits, we observe a sweet spot in which both the average number of Grover iterations and the average number of classical GA generations are minimized. Table 3 summarizes the results presented in Figs. 12B, 12D, and 12F.

## Graph coloring problem

Consider an undirected graph $G = (V, E)$ where $V$ is the set of nodes and $E$ represent the set of edges. We define $C$ as the set of colors. The graph coloring problem is defined as finding the best way of assigning the colors in $C$ to nodes from $V$, such that no two adjacent nodes, $v_i, v_j \in V, e_{ij} \in E$, have the same color ($c(v_i) \neq c(v_j)$). (*Titiloye & Crispin, 2011*) defines the coloring of $G$ as a mapping $c : V \rightarrow E$, such that $c(v_i) \neq c(v_j)$ if $\exists e_{ij} \in E$. The chromatic number of the graph, $\chi(G)$ represents the minimum number of colors that can color the graph $G$.

**Table 2** **The relation between the number of Grover iterations and the number of fixed qubits.** For the Knapsack problem we defined a search space of five qubits, each representing an individual, and a fitness register of eight qubits. We variate the number of fixed individuals from 0 (representing a pure quantum solution) to 5 (representing a classical GA). We performed simulations in which the classical GA is configured to use a non-adaptive mutation, adaptive mutation probability, and adaptive percentage of the mutated genes.

| | Number of Grover iterations | | |
|---|---|---|---|
| **Number of fixed qubits** | **GA with non-adaptive mutation** | **GA with adaptive mutation probability** | **GA with adaptive percentage of the mutated genes** |
| 0 | 11 | 9 | 13 |
| 1 | 9 | 7 | 11 |
| 2 | 4 | 3 | 6 |
| 3 | 4 | 3 | 6 |
| 4 | 2 | 1 | 2 |
| 5 | 2 | 1 | 2 |

**Table 3** **The relation between the average number of Grover iterations and the average number of GA generations.** For the Knapsack problem, we defined a search space of five qubits, each representing an individual, and a fitness register of eight qubits. We varied the number of fixed individuals from 0 (representing a pure quantum solution) to 5 (representing a classical GA). We performed simulations in which the classical GA is configured to use a non-adaptive mutation, adaptive mutation probability, and adaptive percentage of the mutated genes.

| | Average number of Grover iterations | | | | | | Average number of GA generations | | | | | |
|---|---|---|---|---|---|---|---|---|---|---|---|---|
| **Number of fixed qubits** | **0** | **1** | **2** | **3** | **4** | **5** | **0** | **1** | **2** | **3** | **4** | **5** |
| GA with non-adaptive mutation | 11 | 9 | 4 | 4 | 3 | 3 | 0 | 10 | 10 | 10 | 10 | 15 |
| GA with adaptive mutation probability | 9 | 6 | 4 | 4 | 2 | 2 | 0 | 10 | 10 | 11 | 10 | 17 |
| GA with adaptive percentage of the mutated genes | 14 | 11 | 6 | 6 | 3 | 3 | 0 | 10 | 10 | 11 | 11 | 18 |

The graph coloring problem has multiple applications, such as timetabling, scheduling, radiofrequency assignment, computer register allocation, printed circuit board testing, and register allocation (*Mahmoudi & Lotfi, 2015*; *Hennessy & Patterson, 2018*). Others identify applications of graph coloring in routing and wavelength assignment, dichotomy-based constrained encoding, frequency assignment problems, and scheduling (*Demange et al., 2015*; *Orden et al., 2018*).

For the graph coloring problem, an example search space is defined by the graph presented in Fig. 13A. We variate the number of fixed qubits in the individuals' register and perform each simulation 100 times. The classical GA uses non-adaptive mutation with a rate of 0.00002. We use roulette-wheel selection, single-point crossover, and random mutation. The crossover probability is 0.6, with 2 parents involved. We evolve a population

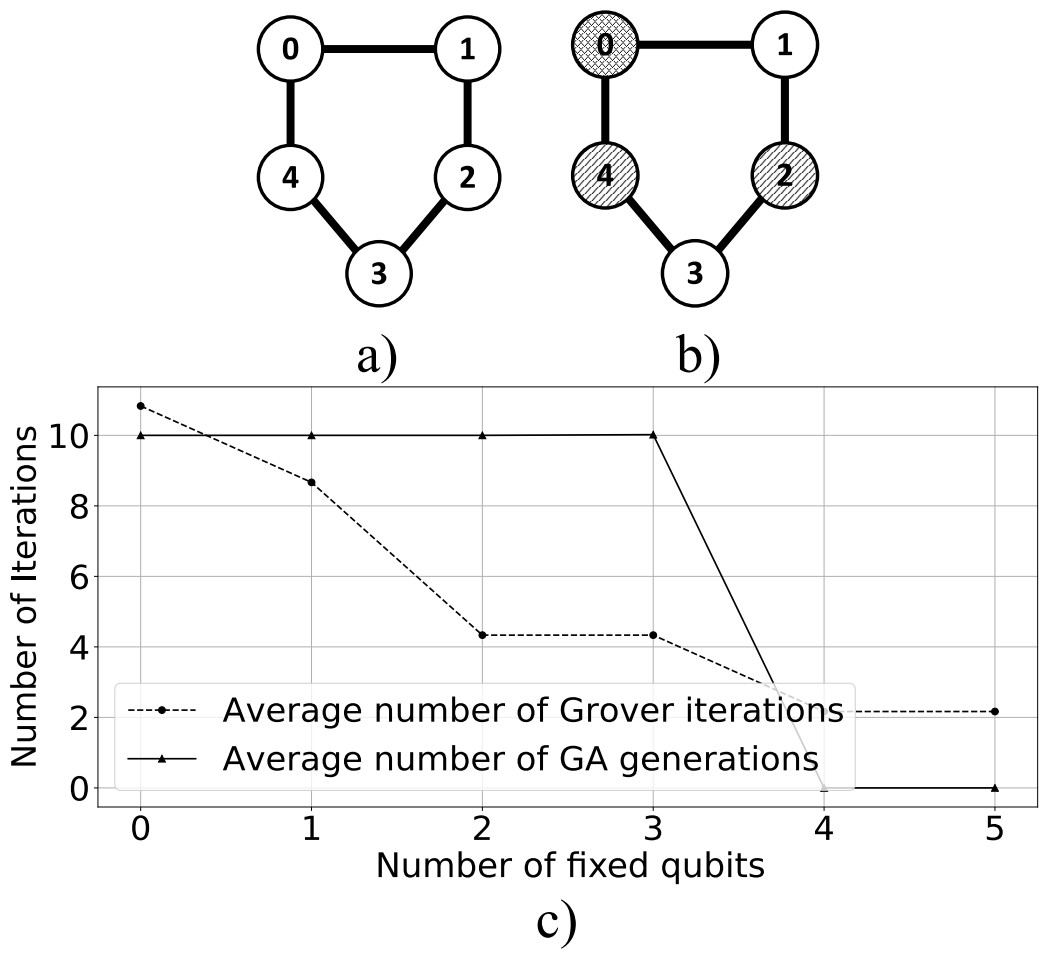

**Figure 13** **(A) and (B) show the Erdös-Rényi graph generated with edge probability 0.7 and 5 nodes, and the solution that colors the graph. (C) Depicts the experimental results; after 3 iterations the algorithm produced 9 valid solutions, of which 2 are the best.** Individual's register size is $n = 10$ and Fitness register size is $m = 8$.

of 100 individuals over 100 generations. As presented in Fig. 13B, the algorithm solves the Graph Coloring problem and determines the chromatic number. In Fig. 13C, we present the relationship between the average number of Grover iterations and the average number of classical GA generations. As observed, the number of Grover Iterations and GA generations decrease as the search space is reduced by fixing genes.

In Supplemental Information, Graph coloring problem, Figs. S19A and S19B we present the graph used for coloring and the solution. In Figs. S20, S21, S22, and S23 we present the outcome of the HQAGO with different numbers of fixed qubits–from 1 fixed gene in Fig. S20 to 4 fixed genes in Fig. S23. In Table 4 we show a summary of the results presented in Supplemental Information, Graph coloring problem.

**Table 4** **Summary of results from the plots presented in the Supplemental Information, Graph coloring problem.** The table shows the number of fixed qubits, GA generations, RQGA generations, best solutions, and valid solutions for different genetic algorithm (GA) configurations.

| | Number of fixed qubits | Number of GA generations | Number of RQGA generations | Number of valid solutions | Number of best solutions |
|---|---|---|---|---|---|
| | 0 fixed individuals (pure quantum solution) (*Ardelean & Udrescu, 2022b*) | 0 | 3 | 8 | 2 |
| Conventional GA with non-adaptive mutation | 1 fixed individual | 10 | 2 | 8 | 3 |
| | 2 fixed individuals | 10 | 2 | 47 | 7 |
| | 3 fixed individuals | 10 | 2 | 45 | 15 |
| | 4 fixed individuals | 10 | 2 | 62 | 10 |

## CONCLUSIONS

This article presents a novel quantum genetic algorithm, based on RQGA, that controls the algorithm complexity by reducing the search space. Accordingly, the proposed HQAGO solves NP-hard problems in $\mathcal{O}\left(\sqrt{2^{n-k}}\right)$ oracle queries.

Therefore, the main advantage of our approach is that it boosts searches large solution spaces using a limited number of qubits. More precisely, compared to the state-of-the-art, our algorithm enables solving complex problems using fewer qubits at the cost of adding extra circuitry to instantiate the conventional GA.

The limitation of our approach is that—from a theoretical standpoint—by fixing $k$ individual's chromosome qubits, the conventional genetic algorithm may exclude the maximum-fitness solution(s). Dealing with such undesired situations may require running the HQAGO several times or optimizing the conventional GA part. Even with an elementary, straightforward approach to designing the conventional GA in this article's simulations, we still obtained the best solutions in all HQAGO runs. Further research on more sophisticated conventional GA methods, which may include combining our approach with similar ones, should lead to even better performance.

The use cases of the HQAGO are the typical application cases for classical GAs, varying from scheduling problems to molecular docking and neural network optimizations. Consequently, HQAGO can be used for register allocation as presented in (*Hennessy & Patterson, 2018*), Wi-Fi channel assignment in *Orden et al. (2018)*, and scheduling applications (*e.g.*, PCBs on a single machine for processing, see *Maimon & Braha (1998)*, scheduling of hard real-time tasks, see *Nawrocki et al. (2009)*). HQAGO can also be used in molecular docking to predict the bound conformations of flexible ligands to macromolecular targets (*Westhead, Clark & Murray, 1997*; *Morris et al., 1998*). Searches performed with HQAGO can be effectively employed in RNA secondary structure prediction since GAs are utilized for the simulation of the RNA folding process and the investigation of possible folding pathways (*Van Batenburg, Gultyaev & Pleij, 1995*). Neural network optimization may also apply HQAGO due to its reduced/controlled algorithm complexity. Indeed, classical GAs are already utilized for the Back-Propagation (BP) algorithm optimization, see *Ding, Su & Yu (2011)*. (As mentioned by the authors, the network trained with GA and BP has better generalization ability and good stabilization

performance.) GAs are also used for tuning the structure and parameters of a neural network to reduce the fully connected neural network to a partially connected network (*Leung et al., 2003*); thus, HQAGO can be beneficial for artificial intelligence applications as well.

In the mentioned use cases, the search space varies between $2^{25}$, as shown in *Nawrocki et al. (2009)*, and $10^{30}$ for the RNA folding as in *Westhead, Clark & Murray (1997)*. In such instances, HQAGO requires runtimes of the orders $\mathcal{O}\left(\sqrt{2^{25}}\right)$, and $\mathcal{O}\left(\sqrt{10^{30}}\right)$. Compared to a fully-quantum solution, the HQAGO's convergence requires fewer generations by marking $k$-qubits and discarding the less-fit individuals and the circuit complexity decreases due to a reduced number of quantum gates and qubits.

**Acronyms**

| | |
|---|---|
| **BP** | Back-Propagation |
| **GA** | Genetic Algorithm |
| **HQAGO** | Hybrid Quantum Algorithm with Genetic Optimization. |
| **QCLAA** | Quantum Carry Look-Ahead Adder |
| **QGA** | Quantum Genetic Algorithm |
| **QGOA** | Quantum Genetic Optimization Algorithm |
| **QIGA** | Quantum-Inspired Genetic Algorithm |
| **QRCA** | Quantum Ripple Carry Adder |
| **RQGA** | Reduced Quantum Genetic Algorithm |

# ACKNOWLEDGEMENTS

We acknowledge the use of the IBM Q for this work. The views expressed are those of the authors and do not reflect the official policy or position of IBM or the IBM Q team.

## Funding
The authors received no funding for this work.

## Competing Interests
The authors declare there are no competing interests.

## Author Contributions
- Sebastian Mihai Ardelean conceived and designed the experiments, performed the experiments, analyzed the data, performed the computation work, prepared figures and/or tables, authored or reviewed drafts of the article, and approved the final draft.
- Mihai Udrescu conceived and designed the experiments, performed the experiments, analyzed the data, performed the computation work, prepared figures and/or tables, authored or reviewed drafts of the article, and approved the final draft.

## Data Availability

The Hybrid quantum search with genetic algorithm optimization is available at Github and Zenodo:

- https://github.com/sebastianardelean/hqago.

- Sebastian Ardelean, & ars. (2024). sebastianardelean/hqago: HQAGO framework (hqago). Zenodo. https://doi.org/10.5281/zenodo.12535319.

## Supplemental Information

Supplemental information for this article can be found online at http://dx.doi.org/10.7717/peerj-cs.2210#supplemental-information.

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
