# Peer review of "Hybrid quantum search with genetic algorithm optimization"

_PeerJ Computer Science, doi:10.7717/peerj-cs.2210_

## Round 0.1 · original submission · Major Revisions

Dear authors,

Thank you for submitting your article. Feedback from the reviewers is now available. It is not recommended that your article be published in its current format. However, we strongly recommend that you address the issues raised by the reviewers, especially those related to readability, experimental design and validity, and resubmit your paper after making the necessary changes. Reviewer 1 has requested that you cite specific references. You may add them if you believe they are especially relevant. However, I do not expect you to include these citations, and if you do not include them, this will not influence my decision

Best wishes,

Reviewer 1 ·

Basic reporting

The manuscript introduces novel elements but requires further refinement before it can be reconsidered for publication. Firstly, the authors need to clearly articulate the significant contributions of their work, as the novelty of the paper is not adequately highlighted. In the introduction, it would be beneficial to include discussions on other contemporary metaheuristic algorithms, such as “A novel optimization approach based on unstructured evolutionary game theory” and “A novel hybrid search strategy for evolutionary fuzzy optimization approach,” to provide a broader context and enhance the manuscript's relevance.

Additionally, the manuscript lacks clarity in demonstrating the advantages and limitations of the proposed approach, especially in comparison to similar methodologies. A more detailed analysis of these aspects would greatly enhance the paper's value and reader comprehension.

Lastly, the overall structure of the paper needs revision. It is advisable to incorporate a few sentences at the beginning of each section outlining its purpose. This structured approach will help readers better understand the logical flow of the paper, facilitating a clearer grasp of how each section contributes to the overall thesis and findings.

Experimental design

The experimental design adheres to established scientific standards, ensuring its robustness and reliability. Consequently, there are no comments in this section as the methodology employed meets the requisite criteria for accuracy and reproducibility in scientific research. This compliance with scientific norms underscores the credibility of the experimental results presented.

Validity of the findings

The validity of the results is significant due to their derivation from the scientific method. This methodological approach ensures that the findings are both reliable and reproducible, providing a sound basis for conclusions. By adhering to rigorous scientific protocols, the study establishes a strong foundation for the credibility and significance of its outcomes.

Additional comments

No comments

Cite this review as

·

Basic reporting

1. Some words need to be corrected. For example - the spelling of the title 'Background' (Page 3) and Fig 6 caption 'controlled'.
2. The entire third paragraph of the 'Results' section is written in some language other than English (Line 270), leading to discontinuity and ambiguity for the readers.
3. The citations need to be appropriately linked to references.
4. References should be rechecked and re-written in proper format. For example- see reference number 3.

Experimental design

The organization of the paper is good. However, these areas can be improved -
1. A small description of the graph coloring and knapsack problem will be good.
2. A paragraph on the motivation behind using QGAs would be helpful in the introduction section.
3. The steps of the RQGA algorithm in the 'Background' section (page 3) can be presented in a pseudo-code format.
4. The derivation and description of the time complexity of the RQGA can be stated in a more precise and in-depth manner.

Validity of the findings

No comments.

Additional comments

The references of the figures, tables, plots, and algorithms are not linked to their original objects. This should be rechecked thoroughly.

Cite this review as

Reviewer 3 ·

Basic reporting

The manuscript is well-written.
Journal template must be followed carefully.

Experimental design

The experimental design shows a good level of thoroughness and precision.

Validity of the findings

The results need to be rewritten.
The problems should be clearly defined before presenting the results.
Numerical tables corresponding to the given figures are missing.
Tables based on the figures must be included.

Annotated reviews are not available for download in order to protect the identity of reviewers who chose to remain anonymous.
Cite this review as

---

## Round 0.2 · accepted · Accept

Dear authors,

Thank you for revising the paper and for clearly addressing the reviewers' comments. I confirm that the paper is improved. Your paper now seems to be acceptable for publication in light of this revision.

Best wishes,

Reviewer 1 ·

Basic reporting

The authors have addressed all my observations. Therefore, the article should be accepted as it is.

Experimental design

The authors have addressed all my observations. Therefore, the article should be accepted as it is.

Validity of the findings

The authors have addressed all my observations. Therefore, the article should be accepted as it is.

Additional comments

The authors have addressed all my observations. Therefore, the article should be accepted as it is.

Cite this review as